# Concept and demonstration of a low-cost compact electron microscope enabled by a photothermionic carbon nanotube cathode

Casimir Kuzyk [1,2], Alexander Dimitrakopoulos [1,2] & Alireza Nojeh [1,2] ✉

The scanning electron microscope (SEM) delivers high resolution, high depth of field, and an image quality as if microscopic objects are seen by the naked eye. This makes it not only a powerful scientific instrument, but a tool inherently applicable to nearly all fields of study and curiosity involving the small scale. However, SEMs have remained complex, expensive, and beyond the reach of many. To broaden access, we demonstrate an SEM using simple, low-cost, off-the-shelf components, and hobby-level electromechanics; this has been enabled by a thermionic electron source based on a carbon nanotube array excited by low optical power. The instrument offers sub-micrometer resolution, a depth of field of the order of a hundred micrometers, and an image quality comparable to commercial SEMs; it also tolerates poor vacuum and moist specimens, making it broadly applicable. It has a flexible design that lends itself to customization for different use scenarios. We describe the conceptual approach and high-level design in this paper; the detailed blueprints of our specific implementation are provided separately online. We hope that specialists and non-specialists alike will build variations that fit their own needs and interests, helping electron microscopy expand further into industry and society.

In 1966, the head of a living flour beetle graced the cover of the San Francisco Chronicle—an example of early microscopic images allowing the public a glimpse into previously unseen intricacies of the small scale[1,2]. Such visuals were obtained with the scanning electron microscope (SEM), taking advantage of its high resolution and depth of field combined with a natural lighting quality. These features combine to produce visuals as if the specimen is seen by the naked eye, but with great magnification. As such, not only has the SEM been a workhorse of high-resolution imaging in industry and academia, but few scientific instruments, aside from the light microscope, have captivated society's imagination as deeply as the SEM. However, unlike light microscopy, which has become increasingly accessible–an example is the $1 US Foldscope[3]–electron microscopy remains inflexible, expensive, and inaccessible to both the public and many professionals. Even those with access to commercial instruments typically have to adapt to the

SEM by preparing a specimen for ex-situ imaging in a shared facility, which precludes many experiments. It would be beneficial if a broader part of society–including researchers, grade-school students[4], healthcare providers in remote regions, ecologists, mining engineers, air and water quality monitoring experts, and artists–had the power of the SEM at their fingertips.

Shortly after the development of charged-particle optics and the first transmission and scanning electron microscopes in the 1920s–1940s, pioneers such as Delong and Oatley ("father of the modern SEM"[5]) envisioned the utility of simpler and lower-performance but more accessible instruments[6–8]. The challenge of enabling broad access to electron microscopy has been undertaken in various forms since. Table-top instruments are now available from established manufacturers and new players, such as the ThermoFisher Scientific Phenom, the Delong Instruments LVEM 5, and the Voxa

[1]Department of Electrical and Computer Engineering, The University of British Columbia, Vancouver, BC, Canada. [2]Quantum Matter Institute, The University of British Columbia, Vancouver, BC, Canada. ✉e-mail: alireza.nojeh@ubc.ca

Mochii™ to name a few, but these instruments typically cost over ~ $50,000 US (and commonly over $100,000 US). Project NanoMi, led by the National Research Council Canada, introduces an open source electron microscope design[9–14]. Miniaturized instruments have also been proposed and relevant components developed to various degrees[15–29] (and commonly tested using the electron beam of a commercial SEM), including in the context of electron-beam lithography and multibeam applications[30–40]. The development of microfabricated components has also seen recent advances as part of a systematic effort at Wrocław University of Science and Technology to create miniature electron microscopes and other devices[41–43]. An alternative path to accessibility is for skilled enthusiasts to build their own SEM–an example is presented on the YouTube channel Applied Science[44]. However, the SEM, as known today, is a complex machine, and it is rare for an individual to undertake the challenge of building one. A different approach, avoiding specialized skills and expensive components, may help open this path broadly.

Here we present the conceptual design and imaging performance of a type of SEM that is simple and built from low-cost, common components using hobby-level prototyping and electronic skills, and which opens a useful region in the performance-cost-flexibility space. The design allows customization for portability and integration into other experimental setups or workflows, helping adapt the SEM to the user.

The detailed design of the specific prototype that we have developed, including mechanical structure, electronic circuits, codes, and components list, is linked in the Data availability section. We note that the concept presented here does not rely on a particular embodiment. Our prototype is only one example, and variations of the instrument can be made using different designs to suit specific application scenarios; the form factor, exact components, control circuits, and software may vary as long as the overall concept remains consistent with what is described here.

## Results

### Approach

A schematic outlining the design concept and main components of the prototype is shown in Fig. 1. The key ingredient is an electron source based on an array of aligned carbon nanotubes (CNTs), also known as a CNT forest, excited by a thumb-size laser[45]. A low-power high-voltage supply, small electromagnets, a permanent ring magnet with no pole-piece, apertures, and generic photodiodes complete the electron-optical column. The source and column are tolerant of poor vacuum. Specialized electron-optical components and engineering are avoided. The structure is built using common machine shop tools and the system is controlled by a basic microcontroller. We note that the photos in the Fig. 1 schematic are of the actual components used

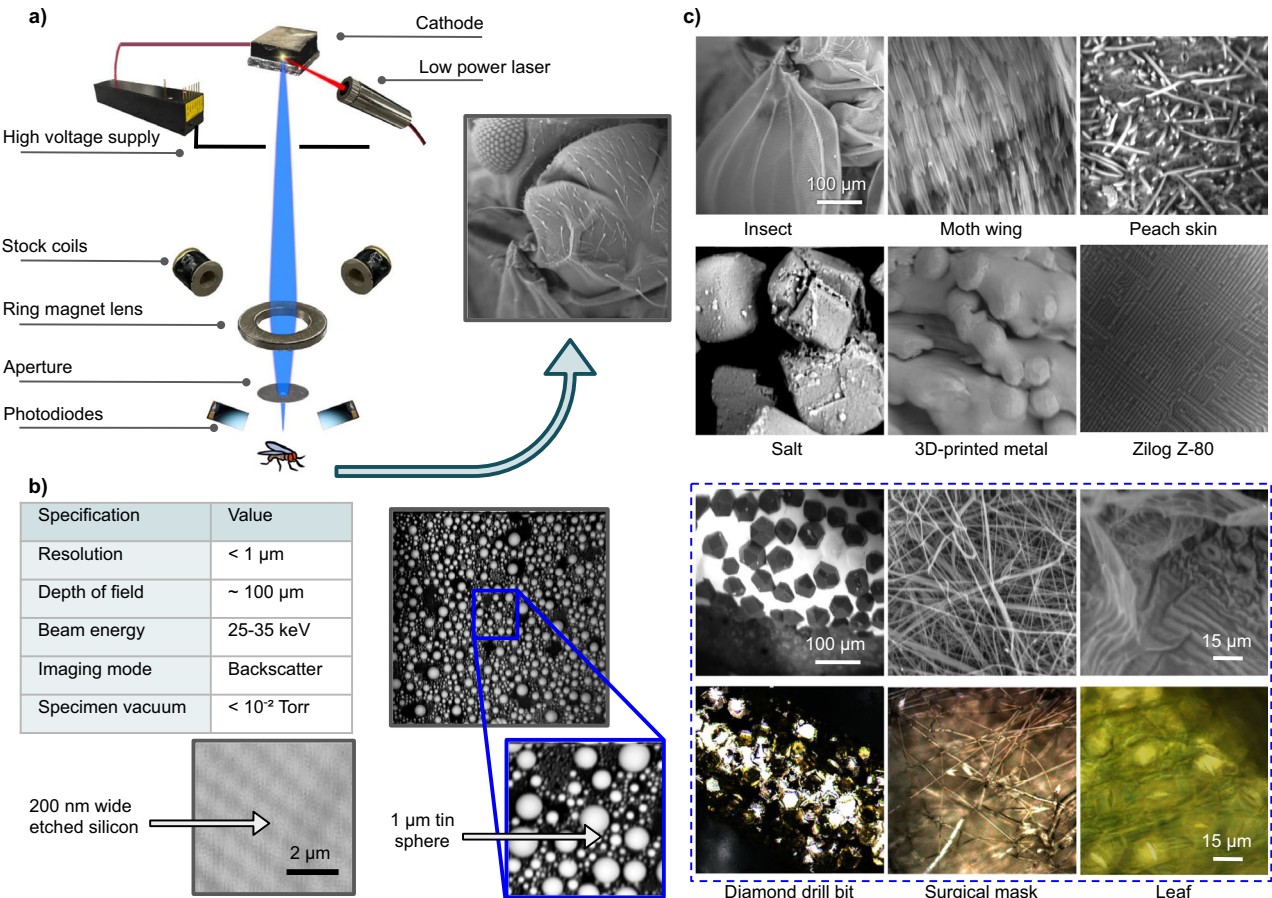

**Fig. 1 | Instrument concept and performance. a** Conceptual schematic of this SEM including actual photos of its key components, all of which are generic and not custom-built for this instrument, and an image of an insect head taken using the prototype. **b** Performance specifications of the instrument and images of two commercial calibration specimens (etched-lines-in-silicon and tin-spheres-on-carbon) revealing this SEM's sub-micrometer imaging resolution. **c** The top three rows include images of diverse specimens obtained using this SEM, with an acceleration voltage between 25 and 35 kV (scale bars approximate). It is worth pointing out that beam stigmation was not used for obtaining these images. (The insect, moth wing, and surgical mask samples were metal-coated before imaging, whereas the others were not). For comparison, the bottom row includes light microscope images of the same specimens as in the third row. The light microscope used was a high-quality, research-grade instrument, yet the complimentary, and in some cases superior, imaging quality of this SEM prototype can be seen, notably its greater depth of field.

in the prototype, and emphasize their simplicity and low-cost nature; this will be discussed in more detail later.

Images of various specimens obtained using backscattered electrons are also shown in Fig. 1. The instrument's large depth of field and intuitive image quality are seen—features that can be better appreciated in a direct imaging comparison with an advanced optical microscope (also in Fig. 1)—demonstrating the capability and usefulness of this SEM.

Also shown in Fig. 1 are a table summarizing the prototype's main specifications, as well as images of a calibration grid and a tin-on-carbon specimen, showing the prototype's resolution of a few hundred nanometers—we claim a conservative value of better than 1 micrometer. This is consistent with the instrument's electron beam probe size of <310 nm as obtained from point-projection imaging experiments, charged-particle optics simulations, and analytical calculations, as will be described later. This imaging performance level, while not competitive with conventional commercial SEMs, fills a gap that is relevant to a range of applications. For example, the effectiveness of SEMs with even relatively low magnification has been established for clinical applications[46].

This SEM was conceived and designed around the above-mentioned CNT forest electron source, which enabled a simplified overall system. This source uses a localized light-induced heating effect[45,47–51]: A spot on the CNT forest is illuminated by a focused beam of light. Due to effectively low thermal conduction loss[52–54], the optically-generated heat remains localized to an area approximately the size of the incident light spot, leading to a strong temperature rise and thermal electron emission (more commonly known as thermionic emission). This can be achieved using modest optical irradiance as readily available from a laser pointer, or even sunlight focused by a handheld lens. The incandescent glow of such a "Heat Trap" spot is depicted on the CNT forest at the top of the Fig. 1 schematic. This electron source combines several advantages of different conventional sources: tolerance of poor vacuum (more tolerant than conventional emitters); not requiring heating power electronics and thermal management (similar to field-emitters); optically-defined electron emission spot (similar to photo-emitters). The combination of these features allows one to simplify the electron beam focusing optics, the structural design, and the vacuum requirements. More detail about the electron source as relevant to this SEM is given in Methods; also see the Carbon nanotube forest growth section in Supplementary Information. (Here we note that nanotube/nanowire-based field-emitters have been studied for various applications, including incorporation into microcolumns and conventional electron microscopes[55–61], and emphasize that the source employed here is not a field-emitter, but an optically-excited thermionic emitter.)

## Instrument design, build, and analysis

A photo of the SEM prototype and cuts of the main sections of its electron-optical column are shown in Fig. 2. The CNT forest is excited by a thumb-size laser and the emitted electrons are accelerated using a low-power supply and travel down the column toward the specimen. Small electromagnets are used for raster-scanning the beam (as well as alignment and stigmation, if desired). A simple N52 neodymium ring magnet, without a pole-piece, acts as the focusing lens; coarse focusing is carried out by adjusting the position of the ring magnet along the column, and fine focusing by tuning the acceleration voltage. Backscattered electrons are detected using basic silicon pn-junction photodiodes, not specifically designed for high-energy electrons, facing the specimen. (It is conceivable that some X-rays are also detected by the diodes and contribute to the image signal together with the backscattered electrons.) The specimen can be moved and repositioned as desired during live imaging at >5 frames/second by manually sliding the plate on which it resides—an experience similar to that of using an ordinary light microscope. The instrument is controlled by a basic microcontroller, which communicates with a graphical user interface written in Python on a laptop computer using a USB connection. The entire SEM was made using hobby-level machining experience and off-the-shelf components. CNT forests are also available commercially, costing a few hundred dollars—a price that can be decreased through bulk production. The design is further detailed in Methods.

The cost of components to build this prototype, not including the vacuum pump (more on that below), was ~$5000 US; this can be cut to almost half if the machining is carried out in-house and/or structural components are made using simpler and additive manufacturing techniques, as well as by using lower-cost alternatives to the high-voltage power supply and feedthrough.

We made two SEM prototypes, which were identical in terms of electron-optical performance, demonstrating the repeatability of the design; the more recent prototype (shown in Fig. 2) incorporated the specimen movement mechanism and updated electronics and system software to allow live imaging.

An overview of the electron-optical analysis of the column is presented in Fig. 3a. The electron emission spot is <40 μm in diameter (as defined by the focused laser spot on the CNT forest surface). The electrostatic field created by the negatively charged cathode cup has a focusing effect on the emitted beam to create a crossover spot with a diameter of <10 μm immediately following the anode aperture. This crossover is then demagnified 50–65 times by the objective lens (depending on the acceleration voltage, which is typically in the range of 25–35 kV), which would yield an electron probe with a diameter of <200 nm at the specimen in absence of aberrations. Chromatic and spherical aberration contributions to the probe size are calculated at 32 nm and 3 nm, respectively. Including these effects, we estimate an overall probe diameter of <250 nm. The crossover spot and probe diameter were also measured through point-projection microscopy experiments (Fig. 3b) to be 6 μm and 310 nm, respectively. The details of the dimensions, simulations, calculations, and experiments are given in Methods. These calculated and measured electron probe diameters at the specimen are consistent with the prototype's imaging resolution of better than 1 micrometer.

## Vacuum considerations

While this SEM can be designed for mounting on an existing high-vacuum or ultra-high-vacuum chamber, those are not a requirement. In our imaging experiments, the specimen compartment was typically held at only ~$1 \times 10^{-2}$ Torr. This tolerance to poor vacuum translates to the instrument's suitability for various types of specimens, including moist and biological ones (Supplementary Information section Vacuum system and figures S1 and S2). For example, we were able to pick fresh tree leaves from outside our laboratory and image them directly without dehydrating or metal coating, while experiencing negligible specimen charging (Supplementary Information figure S2). In fact, the specimen compartment does not have special vacuum requirements, and may be made to operate at variable pressures and even at the level of environmental and wet SEM[62–65].

The electron source and column were typically held at ~$1 \times 10^{-4}$ Torr (partly to prevent arcing) using a vacuum pump, but in principle they do not need active pumping, and may instead be permanently sealed-off under vacuum, similar to a cathode ray tube device. This would be desirable for reducing cost and increasing portability (we have previously demonstrated vacuum-sealed glass vessels incorporating CNT forest electron sources[49,66]; other examples are sealed-off gyrotron and X-ray sources using carbon nanotube cathodes[67,68]). In this scheme, the electrons would exit the column through an electron-transparent membrane and strike the specimen in-situ, be it in high vacuum, low vacuum, or even air. This concept has long been established in atmospheric electron microscopy[69–72]; even atomic scale imaging has been demonstrated at ambient pressure[73]. If

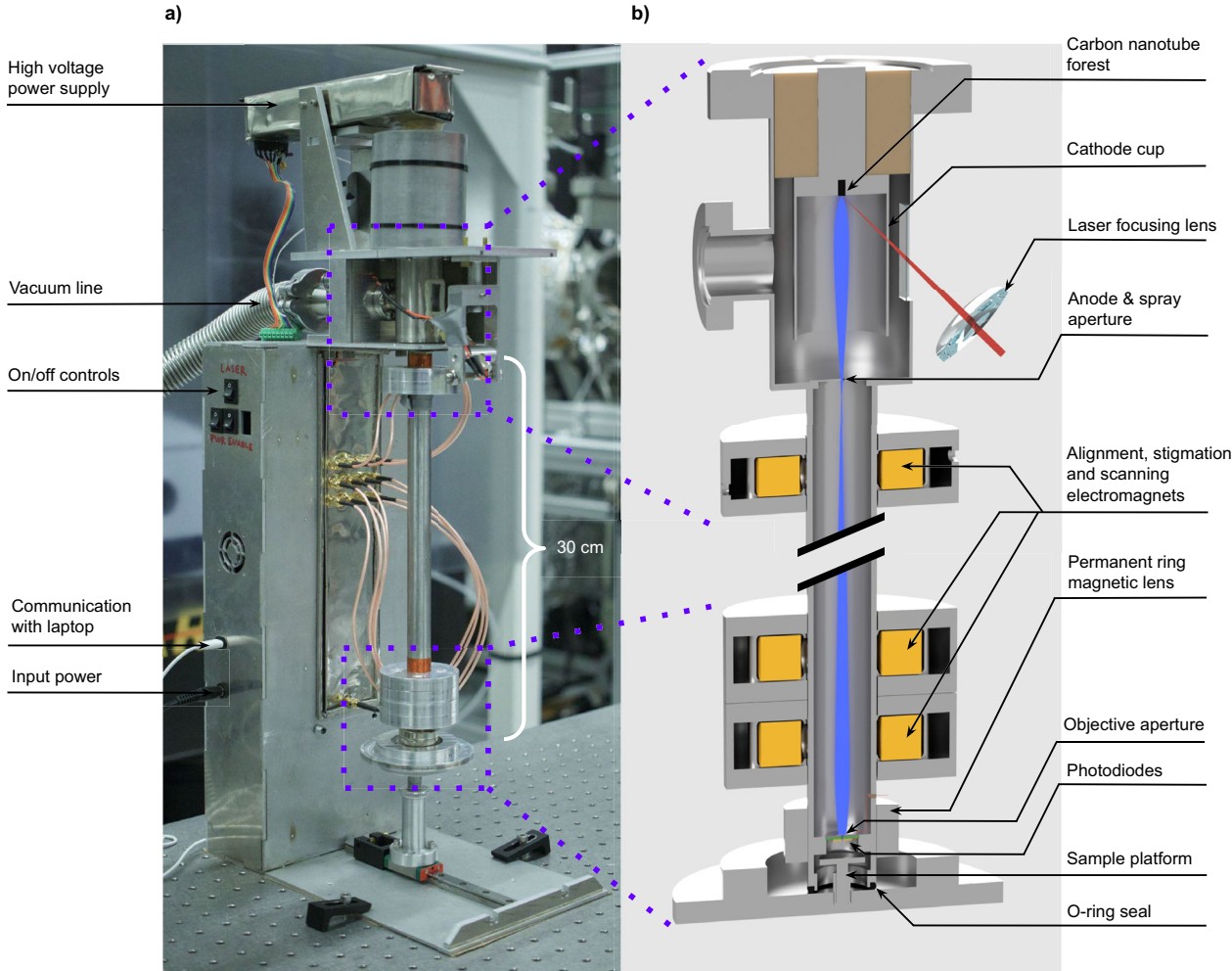

**Fig. 2 | Prototype photo and design. a** Photo of the instrument (laser/X-ray safety covers not shown). Other than the electron-optical column, the rest of the microscope platform is only used to house the electronics and provide mechanical support, and can be replaced by any other desired design/configuration. **b** A computer-aided design section view of the electron-optical column. The detailed description is given in Methods.

the membrane thickness and subsequent travel distance in the specimen environment are within a few electron mean free paths, a useful portion of the beam will remain unscattered and preserve imaging resolution (while the scattered electrons only act to increase background noise). For example, after traversing a 20 nm-thick silicon nitride membrane and 50 μm of atmospheric gas, about 1% of 15 keV electrons are estimated to remain unscattered and, at 0.1 atmosphere, the value rises to 10%; image enhancement methods have also proven effective in this context[74,75]. Below about 0.01 atmosphere, where environmental SEMs operate, scattering in air becomes practically negligible. Scattering in the membrane may be reduced by using thinner membranes made from mechanically stronger materials. For example, bi-layer graphene is shown to improve contrast and signal-to-noise ratio by ~4 and ~12 times, respectively, compared to ~10−20 nm-thick nitride[76]. We have also carried out preliminary demonstrations of pumpless, sealed-off operation, as well as imaging in rough vacuum (~40 Torr) using an electron-transparent silicon nitride membrane; the results are presented in Supplementary Information figure S3.

## Discussion

The premise of this work has been to demonstrate a simple and inexpensive SEM constructed using non-specialist parts and techniques. This instrument delivers images that are qualitatively similar to those of commercial SEMs, and complements or outperforms advanced optical microscopes, covering a range of presently unaddressed applications. It is thus a practically useful instrument that could broaden access to electron-beam imaging in society and industry. Furthermore, it is possible to manufacture sealed-off, pumpless versions of this SEM using the technologies established for cathode ray tube television sets, including their high-voltage supply and control electronics. Large-scale production and distribution at a cost of less than a thousand dollars thus appears conceivable, extending access widely. Taking a broader view than only imaging applications, one may speculate that the availability of a simple point-and-shoot electron beam device, which is the core of this instrument, may prove useful and enable other applications.

In future designs, multiple independent detectors may be placed around the objective aperture to cover different backscattering angular ranges, and their signals used for enhanced topographical imaging. In addition to backscattered electron detection, other established imaging modalities may be used, including electron-beam-induced current and secondary electron detection, for example using gaseous detectors[63,65,77]. Additional electron-optics may also be incorporated to provide magnification after the specimen for transmission-mode imaging. The addition of X-ray or cathodoluminescence detectors would also enable spectroscopy. Furthermore, while the nanotube photothermionic cathode has enabled the present SEM, other electron sources may also be

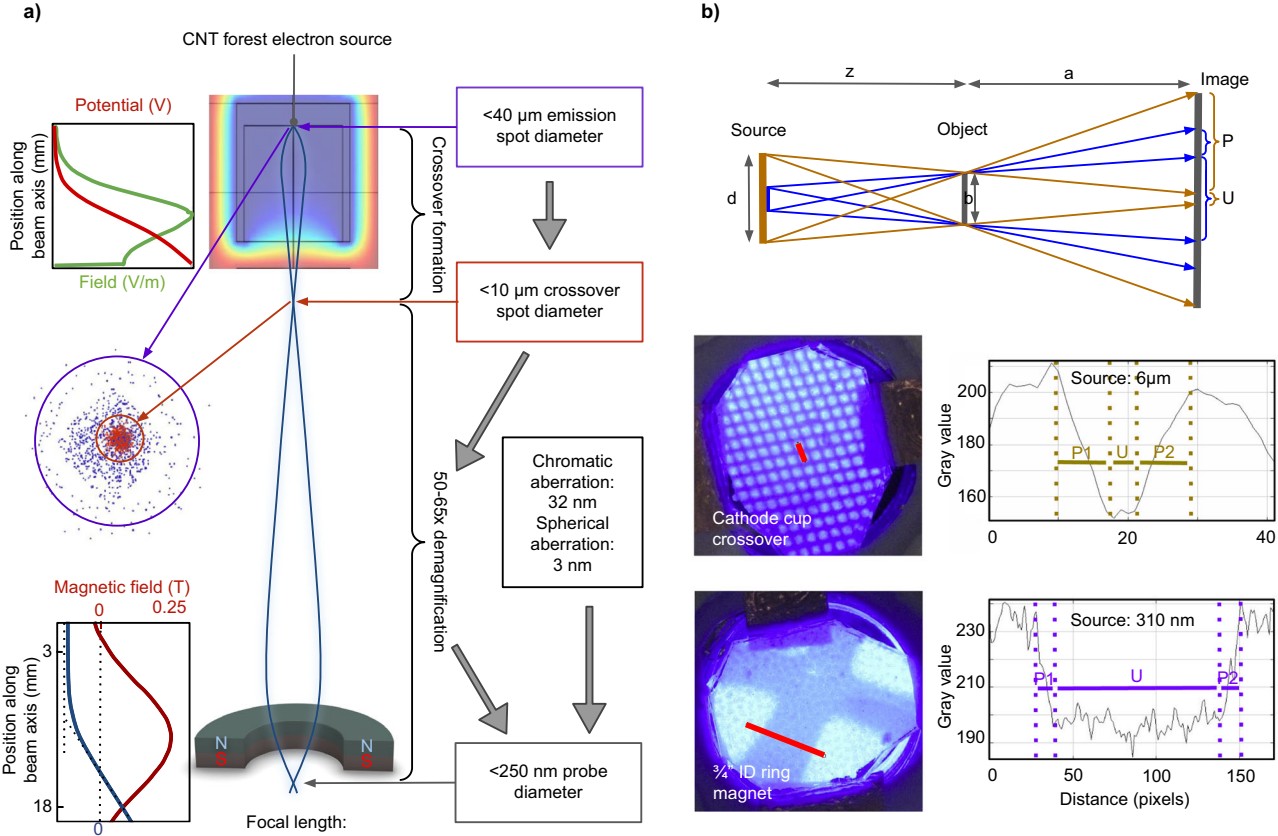

**Fig. 3 | Electron-optical analysis. a** Probe diameter analysis: starting from the electron emission spot diameter of <40 μm from the CNT forest, charged-particle tracing simulations show the formation of a beam crossover with a diameter of <10 μm shortly after the acceleration region (colour map shows the simulated potential distribution, and blue and red dots are beam particles at emission and crossover planes, respectively), which is then demagnified by the objective lens to <200 nm as calculated using beam tracing within the measured axial magnetic flux density distribution (bottom plot) of the ring magnet. Given contributions from spherical and chromatic aberrations, the probe diameter is estimated to be <250 nm. More details on the aberrations and ring magnet focal length calculations can be found in Methods. **b** Probe diameter measurement: (top row) schematic of the point-projection microscopy experiment–we used a metallic 2000 mesh as the object; (middle row) image on the phosphor screen and line profile of the image along the red line for the experiment where the beam crossover is used directly as the source; (bottom row) results for the case where the electron probe (beam crossover after focusing by the ring magnet) is used as the source. The crossover and probe diameters are thus obtained to be 6 μm and 310 nm, respectively. (ID inner diameter; P penumbra; U umbra) The details of these calculations and experiments are given in Methods.

considered for similar instrument designs; low-work function nanowires will be particularly interesting in this regard.

The images shown here were obtained using neodymium ring magnets as the only focusing element. We present additional results using various such magnets, and discuss the use of a pole-piece, a condenser lens, and stigmation for resolution improvement, in Supplementary Information section Electron-optical system and figures S4 and S5. The effect of environmental noise is presented in Supplementary Information section Electromagnetic noise and shielding and figure S6. However, there is room for performance enhancement even without improving the hardware, through software-based filtering of known electrical and mechanical noise, and deconvolution of the point spread function of the electron probe[78,79].

Artificial intelligence (AI) methods have found their way into electron microscopy and related techniques. Recent examples include resolution improvement in three-dimensional imaging[80,81], automating image acquisition and segmentation[82] and crystal structure identification[83], increasing access to atomic-scale dynamics in in-situ microscopy[84], and automating experimental workflows[85]. The use of AI in microscopy is only expected to grow and, going forward, significant progress may originate from innovation in software. The availability of vast image datasets from diverse application domains for both supervised and unsupervised training of AI systems will be important in this regard. Not only will the electron microscopy platform presented here benefit from AI-based resolution enhancement and related advances, but it may also enable large user groups to contribute to diverse image datasets, in turn helping advance the use of AI in electron microscopy. Just as AI models have recently surprised all by their abilities to glean and synthesize information, they may similarly surprise us by their capacity to construct physically legitimate high-resolution electron micrographs based on low-resolution hardware data. Placing SEMs in the hands of many may thus help accelerate the advancement of electron microscopy to make the small world more accessible.

## Methods

### Optically excited thermionic electron source based on CNT forest

As mentioned in the main text, heat remains localized at the illuminated spot on a CNT forest[45,47]–we have characterized this heat localization in detail using both simulations and thermographic and hyperspectral temperature mapping experiments[52,86]. As a result, the thermionic electron emission spot is defined by the focused spot of the illuminating laser beam.

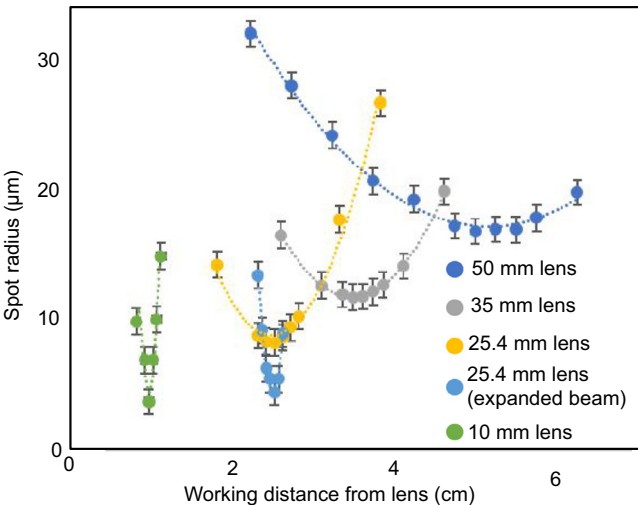

**Fig. 4 | Choice of optical lens.** Focused laser spot radius as a function of working distance for various lenses with different focal lengths; the SEM prototypes used the 50 mm lens, resulting in a focused laser spot radius of <20 μm. The error bars arise from optical power measurement uncertainty in the knife-edge experiments.

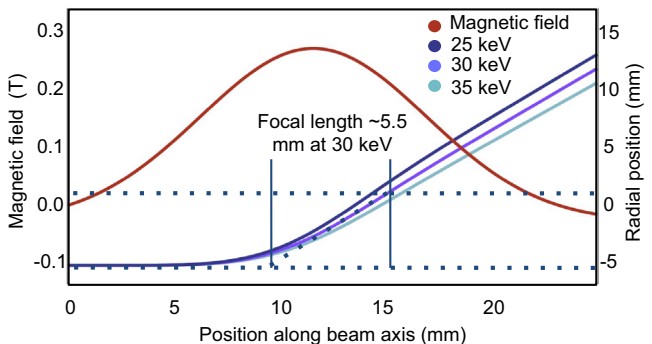

**Fig. 5 | Electron beam focusing using permanent magnet objective lens.** Axial magnetic flux density distribution (Gaussian fit to the measurement data is shown) of the ring magnet objective lens and simulated rays for beam energies of 25, 30, and 35 keV, giving a focal length of -5.5 mm at 30 keV.

In the present SEM, the CNT forest is illuminated by a thumb-size laser whose beam is collimated by its integrated optics and then focused using a plano-convex lens with a focal length of 50 mm. The laser we used had a wavelength of 650 nm and was chosen for its low cost and small form factor. A power of a few tens of mW is sufficient for producing the necessary electron emission current, and the choice of wavelength is a matter of availability; we have experimentally demonstrated this thermionic emission effect with wavelengths ranging from 266 nm to 1064 nm[48,50].

Typical values of emission current are in the range of 1–10 μA and, with most of the electrons being blocked by the apertures, the probe current reaching the specimen is estimated to be on the order of a nanoampere or less. The reduced brightness of the emitter was estimated at $1.82 \times 10^4$ Am$^{-2}$sr$^{-1}$V$^{-1}$ through experiments where the electrons emitted from the CNT forest were accelerated towards a phosphor screen directly facing it[87]. This is comparable to the value reported for a tungsten thermionic emitter[88]. However, this value could be increased by an increase in laser power (and therefore temperature and emission current density) and a decrease in laser spot size.

## Cathode assembly

The electron source assembly consists of a CNT forest attached at the base of the cathode cup (a conductive cylinder similar to that discussed in ref. 89). Other than having a focusing effect on the electron beam to create a crossover point (Fig. 3), the cup also acts to limit the local electric field near the CNT forest and prevent spurious field-emission. The cathode cup length, diameter, and distance to the anode were tuned using charged-particle optics simulations in COMSOL Multiphysics® to minimize the crossover spot diameter, before experimental implementation. The CNT forest and cathode cup are biased at the negative acceleration voltage using a feedthrough directly connected to the output of a negative-polarity variable-voltage supply (capable of up to 40 kV).

We verified in charged-particle simulations and experimentally that, even if the laser spot was not centered on the axis of the beam column, the electric field between the cathode cup and the anode would naturally center the electron beam. As such, alignment of the cathode cup to the beam column axis is more important than alignment of the actual emission spot. The prototype uses parts made in a student metal shop with tolerances greater than +/-0.005", which has been sufficient for mechanical alignment of the cathode cup to the beam column.

Based on knife-edge measurements (Fig. 4), we obtain a focused laser beam waist diameter of <40 μm, with a workable depth of focus of a few millimeters. In the SEM, the laser is focused through a glass window on the cathode vacuum compartment and a small opening on the side of the cathode cup onto the CNT forest (Fig. 2). The glass window is attached to the cathode compartment using low-vapour-pressure epoxy. The laser is held in place in a two-axis movable joint to facilitate its positioning on the CNT forest side surface. The opening on the side of the cathode cup was simulated in COMSOL Multiphysics® to have a negligible impact on the electric field distribution inside the cup, and so does not affect the shape of the electron beam.

Due to the bulk nature of the CNT forest, and the ability to use an emission spot misaligned to the electron-optical axis, the focused optical spot position can be changed periodically. Thus, although the CNT forest sidewall may suffer degradation over time due to high-temperature operation, the lifetime of the emitter as a whole is a collective of all spots' independent lifetimes. We have used the same CNT forest source for 300+ hours of operation without noticeable decrease in imaging performance.

## Analysis of electron-optical demagnification, aberrations, and probe size

Electrons emitted from the cathode accelerate towards the anode. A significant portion of them passes through the 1 mm-diameter anode aperture and travels down the field-free column until the objective aperture, which truncates the beam prior to focusing. Focusing is accomplished by a permanent-magnet objective lens. The magnetic flux density distribution of this ring magnet was measured using a gaussmeter mounted on a translation stage, and fed to a simulation solving the paraxial ray equation, yielding a focal length of 4.6–6 mm for beam energies in the range of 25–35 keV (Fig. 5). Given the distance from the cathode cup crossover plane to the objective lens, the demagnification is 50–65 times. With an initial crossover spot diameter of <10 μm as described before, the probe size is thus <200 nm if aberrations are neglected. Based on the diameter of the objective aperture and its distance to the focal plane, spherical aberration is obtained to be 3 nm using

$$d_{\text{sph}} = 0.5 C_s \beta^3 \tag{1}$$

where $d_{\text{sph}}$ is the spherical aberration, $C_s$ is the spherical aberration coefficient, and $\beta$ is the semi-angle of electron beam convergence. We

have measured the energy spread of thermionic emission from the CNT forest to be <1 eV[90,91], so the ripple of the power supply, which is 9 $V_{p-p}$ at 30 kV, is the dominant source of chromatic aberration, which is obtained to be 32 nm using

$$d_{chr} = 2C_c(\Delta E/E_o)\beta \tag{2}$$

where $d_{chr}$ is the chromatic aberration, $C_c$ is the chromatic aberration coefficient, ΔE is the energy spread of the electron beam, and $E_o$ is the acceleration voltage. We used the focal length of the objective lens as the values of $C_c$ and $C_s$. Given the above demagnification analysis and aberration calculations, we expect the overall probe diameter to be <250 nm.

## Point-projection measurements of the crossover and probe

Point-projection experiments were used to measure the electron beam crossover spot size (due to the focusing effect of the cathode cup) and probe size (due to the ring magnet objective lens). The schematic in Fig. 3b depicts the experimental setup, where the source is either the beam crossover or the probe (depending on the experimental configuration employed), the object is a 5 μm-width bar (from a metallic 2000 mesh), and the image is projected onto a phosphor screen. There are two scenarios in this schematic: one where the source is smaller than the width of the bar, and the other where it is larger. By measuring the umbra and penumbra of the resulting images, we can obtain the source size. Using geometry, the umbra and penumbra for the two scenarios outlined in Fig. 3b can be shown to be

$$d < b : \quad U = b + \frac{a(b-d)}{z} \tag{3}$$
$$P = \frac{ad}{z}$$

$$d > b : \quad U = b - \frac{a(d-b)}{z} \tag{4}$$
$$P = \frac{ad}{z},$$

where $U$ is the width of the umbra, $b$ is the width of the object, $a$ is the distance from the object to the image, $d$ is the width of the source, $z$ is the distance from the source to the object, and $P$ is the width of the penumbra. The distance between the source and the object is obtained based on the magnification seen in the images:

$$M = \frac{(z+a)}{z}, \tag{5}$$

where $M$ is the magnification. Since the projection images using the crossover as a source have a slightly non-zero umbra width, we can assume that the crossover width is comparable to that of the mesh bar, so we can use equation (4) to calculate an upper bound of the source size that would result in an umbra equal to 0. This puts an upper bound of 6 μm on the crossover diameter.

In the case of point-projection experiments using the probe formed by the objective lens as the source, the source diameter is much smaller than that of the mesh bar width given the small penumbra-to-umbra ratio seen in Fig. 3b. Using the system of equations (3) from above, we have two unknowns: the diameter of the source, and the distance from the source to the object. Taking the penumbra as the average of the two values gathered from the line plot in Fig. 3b (bottom row) along with the width of the umbra from the same figure, we find the distance from the source to the object to be 0.54 mm, and the diameter of the source to be 310 nm. Given that the calculations do not account for coulomb interactions in the electron beam, lensing effects from the mesh, or phosphor non-idealities, this measurement result is consistent with the above-calculated probe size of <250 nm and, indeed, with our observed SEM imaging resolution.

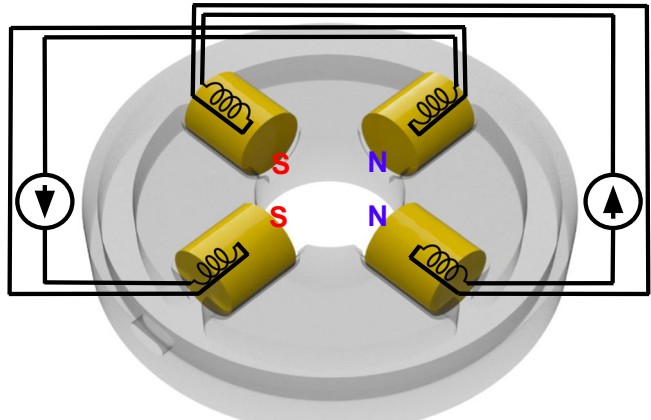

**Fig. 6 | Electromagnetic beam control.** The 4-electromagnet configuration wired for alignment and scanning functions. For the stigmator, the wiring is such that the same poles would be diagonally facing.

## Beam column

The main body of the SEM consists of a 304 stainless steel pipe with a length of 30 cm and inner and outer diameters of 12.70 mm and 19.05 mm, respectively, welded to the bottom of the cathode compartment. The anode is an aluminum disc with a 1 mm spray aperture positioned at the top end of the column, 11.5 mm below the open end of the cathode cup, and is electrically connected to the rest of the column, which is grounded. A set of 4 small, 450-turn electromagnets (with opposing pairs driven by the same current) enables transverse beam alignment (Fig. 6). Near the bottom of the column is a standard 100 μm-diameter objective aperture (made of molybdenum). Immediately below the aperture reside two common silicon pn-junction diodes which act as backscattered electron detectors. Their back sides (cathodes) are soldered onto a small printed circuit board, which is attached to the aperture holder disc and grounded, and their front sides (anodes) are soldered onto a wire that is brought out of the column through a small gap on the side; the gap is sealed around this feedthrough wire using low-vapour-pressure epoxy. Two sets of 4 electromagnets, identical to those used for alignment and wired for the appropriate arrangement of polarities, form the stigmators and scan coils, respectively (however, it should be noted again that the stigmators were not used for the specimen images shown in Fig. 1). The objective lens consists of two axially-magnetized neodymium ring magnets with inner and outer diameters of 19.05 mm and 31.75 mm, respectively, attached to one another in series, without any iron core. The use of permanent magnets to focus an electron beam has been studied in the past[19,20,92].

The bottom end of the column rests on the flat surface of an aluminum disc which holds the specimen, with an elastomer o-ring between the two (Fig. 2). A small amount of vacuum grease is applied to the o-ring to enable the specimen holder disc to easily slide underneath the column while the system is under vacuum. This sliding movement, which is controlled by hand, forms the specimen movement mechanism. The instrument frame, which holds the column and houses the electronics, is made of aluminum.

## Electronics

The electronics consist of drive circuitry for the laser, high-voltage supply, and electromagnets, and a transimpedance amplifier for the detectors. The scan driver control and detector signal acquisition are handled using a basic microcontroller, which communicates with the computer running the user interface. The laser beam, high-voltage supply, alignment coils, and detector are switched on and adjusted by

manual switches and potentiometers mounted on the aluminum box housing the electronic circuits. The instrument is powered by a single 24 V AC-DC wall adapter, and uses less than 50 W of total power.

## Detailed design and components list

The detailed design of the prototype, including mechanical structure, electronic circuits, codes, and components list, is provided separately online and linked in the Data availability section; we also refer the reader to the Safety section of Supplementary Information.

## Pumping system used for the prototype

The prototype was pumped down using a turbomolecular pumping station through a port on the side of the cathode compartment. While the turbomolecular pump can produce ultra-high vacuum (UHV) in an appropriately sealed vacuum chamber, we did not use UHV. In fact, our prototype is far from a UHV-compatible chamber: it uses a non-UHV connection to the pump; an elastomer o-ring and vacuum grease loosely seal the specimen stage to the column allowing for relative movement; several other sealing points in the column and cathode compartment are made by manually applying epoxy; PCB parts and regular solder are used for mounting the detector within the column; and the system is routinely vented to atmosphere and pumped down without baking. Moreover, since the vacuum line is connected to the cathode compartment, the column is evacuated only through small openings, and the specimen compartment is evacuated through even narrower paths such as the objective aperture. These pose significant resistance to flow, hence the poor vacuum levels in the cathode and specimen compartments as stated in the main text. In other words, such a turbomolecular pump is far more capable than needed by this SEM, but was a convenient option for our systematic experiments.

## Data availability

The data that support the findings of this study are available within the paper and its Supplementary Information file. While the work is protected by patents for commercial use, it can be used freely under a license for non-commercial academic research and/or educational purposes. The license and detailed design documentation can be accessed through https://doi.org/10.5281/zenodo.15777424 [10.5281/zenodo.15777424].

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

## Acknowledgements

We are grateful to R. Fabian Pease for insightful discussions and consultations. We also thank Alex Anees for helping develop Figs. 1a and 2a and Gabriel Robinson-Leith for help with an early version of the software.

We acknowledge funding from the Natural Sciences and Engineering Research Council of Canada (NSERC Grants No. RGPIN-2017-04608 (AN), RGPAS-2017-507958 (AN), I2IPJ538549-19 (AN), I2IPJ548806-20 (AN), RGPIN-2023-05154 (AN)) and The University of British Columbia Work Learn program (AN). Laboratory infrastructure was funded through past grants from NSERC (Research Tools and Instruments (AN)), the Canada Foundation for Innovation (CFI Leaders Opportunity Fund (AN)), and the British Columbia Knowledge Development Fund (BCKDF (AN)). This research was undertaken thanks in part to funding from the Canada First Research Excellence Fund, Quantum Materials and Future Technologies Program (CFREF (Quantum Matter Institute)). We acknowledge CMC Microsystems for the provision of access to COMSOL Multiphysics®.

## Author contributions

A.N. conceived the microscope based on the Heat Trap photothermionic electron source. C.K., A.N., and A.D. carried out characterization experiments and designed, built, and operated the first prototype. A.N. and A.D. designed, built, and operated the second prototype. A.N., C.K., and A.D. wrote the manuscript. A.N. supervised the project.

## Competing interests

A.N. is a co-inventor on US patent 10,741,352, Canadian patent 3,048,303, and Chinese patent 110,199,374 covering content in this manuscript, and US patent 9,859,097 and further provisional patents relevant to this work. The remaining authors declare no competing interests.
