## [Transparent Peer Review file · Nature Communications]

Concept and Demonstration of a Low-cost Compact Electron Microscope Enabled by a Photothermionic Carbon Nanotube Cathode

Corresponding Author: Professor Alireza Nojeh

Version 0:

Reviewer comments:

Reviewer #1

(Remarks to the Author)

Democratizing the Electron Microscope
by Casimir Kuzyk, Alexander Dimitrakopoulos, Alireza Nojeh

The manuscript introduces a low cost scanning electron microscope developed by the authors and demonstrates applications.

The manuscript is well written, and can be impactful by making scanning electron microscopy more accessible. The authors make an excellent point that democratizing scientific instrumentation benefits society.

That said, the work presented in the manuscript is not a good match for the Nature Communications in terms of novelty, depth and mainly the topic itself (cheap desktop SEM capable of handling wet samples has been described before.). Therefore, suggest rejecting the manuscript.

Moreover, the manuscript does not provide sufficient information (blueprints, electronics parts list and source code for of their software) to reproduce their implementation of low-cost SEM decreasing the usefulness of the current version of the manuscript.

A more detailed feedback follows below.

While I'm highly supportive of open science and in particular new open hardware for electron microscopy, I believe the manuscript is better suited for publication in a microscopy and / or hardware oriented journal. Example journals could be:

Hardware-X

<https://www.hardware-x.com/content/aims>

or

Methods in Microscopy

<https://www.degruyter.com/journal/key/mim/html?lang=en>

or

Micron

<https://www.sciencedirect.com/journal/micron>

To make their argument on democratizing electron microscopy solid, the authors should publish the blueprints and all the code for their set-up in an accessible manner. The above-mentioned journals as well as Open Science Foundation, GitHub or GitLab are examples of suitable platforms to make the code and blueprints available.

Their claim of sub-5000 dollar part list is misleading. The vacuum pump and controller that they use alone is close to 10,000

dollars.

The use of permanent magnet lens has been well established since 1950's. For example, RCA offered a commercial 50 kV TEM utilizing permanent magnet lens.

http://www.smecc.org/rca_emt_tabletop.htm

More recently, permanent magnet excited lens are being used in Hitachi H300 (1980's), and in TM-1000 and TM-2000 SEMs.

Abstract:

The authors say that their design is a flexible design, suitable for customization. To support such claim they should show some flexibility (other than lens positioning) that their design enables.

Main text:

It is not entirely correct that electron microscopy is not accessible. I'm aware of activities where table-top SEMs are loaned to high-schools free of charge to promote science and microscopy. The program I happen to be aware of has been running close to a decade. So while the authors make it possible to built own SEM, it is not true that SEMs are not accessible.

Projects comparable to the design reported here can be found even on YouTube

<https://youtu.be/VdjYVF4a6iU?si=dgfWAG6QxwaFou7C>

arguably doing a good job democratizing SEM, as pointed out by the authors. In my view the Applied Science YT video provides more useful and detailed practical information than the current manuscript.

In addition to the SEMs mentioned on line 45, both Hitachi and JEOL supply competitive tabletop SEMs.

The authors are not entirely correct in their description of the NanoMi project. It provides detailed information, including blueprints, to build pretty much anything: from physicist-instrument to portable hobby-level tools. It is actually rather simple platform to implement in its basic configuration and detailed information, is available either on Git or upon request.

The authors point the large depth of field in their instrument. However, that feature is inherent to SEM imaging at modest resolution. The application images in Figure 1 are a good choice demonstrating the utility of an low-cost instrument.

The use of laser-heated thermal source is an excellent idea that could be of interest to open-source electron microscopy community due to its practical advantages. The authors have built two instruments, demonstrating the repeatability of their design. It would be desirable to also demonstrate that their design can be also reproduced at other labs.

On line 154 the authors state:

"Chromatic and spherical aberrations are calculated at 32 nm and 3 nm, respectively"

do they mean chromatic and spherical aberration coefficients ? Also check Figure 3 for units.

If so, the units are likely millimetres, not nanometers.

The authors claim that their instrument cost is ~ 5 kU\$. Does it include vacuum pumps ? The text does not seem to specify the type of pumps used, but that is important information.

The summary and outlook section is largely speculative and not supported by results

"Multiple independent detectors may be placed around the objective aperture to cover different backscattering angular ranges, and their signals used for enhanced topographical imaging. "" has not been demonstrated here.

"In addition to backscattered electron detection, other established imaging modalities may be used, including electron-beam-induced current and secondary electron detection using simple gaseous detectors. Additional electron-optics may also be incorporated to provide magnification after the specimen for transmission-mode imaging.""

While all of this is plausible, it was not demonstrated and can not be claimed.

"" The addition of X-ray or cathodoluminescence detectors would also enable spectroscopy."

As above, when not demonstrated it should not be claimed in a journal with level of Nature Communications.

While AI is an important direction in electron microscopy, there is no link, beyond speculative suggestions, between developments and the information in the manuscript.

--- Methods:

The use of laser-illuminated CNT forest as a thermal emitter is arguably a novel feature. In any case, it is an important aspect

making the instrument robust to poor vacuum. As such, the authors should provide sufficient detail to enable others easily reproduce their set-up. For example, the (commercial) source of CNT samples, the model of the laser(s) tested etc. should all be provided.

The authors should provide a plot of electron beam current versus time and possibly illumination power to support their claim of the electron current provided by the source and of its stability.

"... " A set of 4 small, 450-turn off-the-shelf ... "

and

"The objective lens consists of two off-the-shelf, axially-magnetized neodymium ring"

The authors need to specify in more detail what they used. When they mention "off the shelf" component, it is necessary to include manufacturer and model.

Values of figures such as Fig 3 is nearly zero (they exist in nearly all textbooks of electron microscopy) unless they provide detail of a particular implementation, such as dimensions, coil type, exciting currents etc.

Paragraph "electronics" is useless. There is no information beyond the irrelevant fact that the authors like Teensy microcontroller and do not need much power.

As for the vacuum section, the controller and TMP the authors use costs in the order of 10,000 dollars. This implies that their stated 5,000 dollar instrument cost is misleading. Furthermore, even with Quik Flange connectors and correct (e.g. Torr Seal) epoxy and multiple moving parts, the authors should be able to routinely achieve 10^{-6} torr or less.

In the supplement. info the authors again refrain from including useful information, such as thickness and shape of mu metal shielding, type and dimensions of magnets used for objective lens etc. While I do agree that the optics performance is and choice of magnets is not critical, the authors need to include such information to make it possible for others to reproduce their design.

Reviewer #2

(Remarks to the Author)

The article presents an innovative method for making a scanning electron microscope from simple, inexpensive and available materials and elements. According to the authors, the price of the components and manufacturing techniques used does not exceed USD 5,000 (excluding the vacuum pump), which is an impressive result when it comes to the price of the manufactured device. The presented operating parameters of the microscope are not as good as those of its expensive counterparts, but competing with high-tech devices, whose design has been developed for decades, is basically impossible. The image resolution presented in the article is much better than that of the best optical microscopes. This was the goal of the creators, to provide users with an electron microscope that is cheap to manufacture and easy to use, and at the same time its imaging capabilities are below 1 μm , thanks to which it can be successfully used to diagnose sub-micrometer structures without additional preparation. This is detailed in the manuscript. The possibility of using the microscope without complicated pumping systems and in an average vacuum is an additional advantage that reduces the costs of such a device (although the purchase of the turbomolecular pump used in the presented set is a large expense), especially since the authors show that it is possible to operate such a device without a vacuum pump, as a tightly closed vacuum device, but this requires additional research and technological work.

The presented work is very important for the field of vacuum and electron beam techniques, such as scanning electron microscopy. Until now, the development of modern microscopic techniques has taken place in closed laboratories of companies that produce these devices. The demonstrated invention provides tools for the development of scanning electron microscopy in the comfort of your home or in any laboratory that would like to have an electron microscope. The authors describe in detail the steps needed to produce this device and also show the latest trends in the development of electron microscopy, in the form of work leading to the miniaturization of electron microscopes or their use for observing samples placed in atmospheric pressure. They list the possibilities of developing their invention, which can be implemented by other research teams and possibly also by the creators themselves. The presented work shows that the development of even advanced devices can be carried out in any laboratory and that there is still a need for innovations in the field of science and technology.

The conclusions of this article are supported by many images of different samples taken in the developed microscope, as well as by detailed analyses of the electron path in the developed electron-optical column. The possibilities of focusing the electron beam using a permanent magnet and further possibilities of developing the electron-optical column are shown. The work of the carbon nanotube emitter is discussed, and sources where more detailed information about the electron source can be found are given. The article explains that the work of such a source does not require high or ultra-high vacuum, and its operation time is very long, which also confirms the usefulness of the technology used for its invention.

All data and analysis are insightful and appear to be accurate. The analysis is well thought out, and the results are presented clearly. No changes are required in this area.

The article presents the production of a scanning electron microscope under home conditions without the need for knowledge of advanced engineering techniques. For this reason, the presented methodology is simplified to the minimum and is described in such a way that it is possible to produce a similar device on your own. The authors themselves produced

a forest of carbon nanotubes that served as a source of electrons, which would be difficult to do at home. However, they write that such structures are available for sale, but searching the Internet, it is difficult to find examples of companies selling such products, so it would be worth adding a link to an example company to confirm that such products are actually available. In summary, this article may be a breakthrough in the availability of scanning electron microscopes for scientists and people who want to use the benefits of these machines in their research. Since I did not find any major flaws in the article, I suggest that it be approved for publication.

Reviewer #3

(Remarks to the Author)

Congratulations to the authors on their fantastic expectations to have a Scanning Electron Microscope (SEM) at a cost of around a thousand dollars. However, it seems that in my opinion it is too greedy to have a sealed column SEM with 25 to 35KeV electron beam energy at that cost and the mentioned column size. I propose that you can attempt to have an SEM with a beam energy between 1 to 5KeV at a cost as you wish. Please avoid using aluminum apertures.

DETAILED RESPONSES TO REVIEWERS' COMMENTS

We are grateful to the reviewers for their time and feedback. Below we present point-by-point responses to the reviewers' comments and indicate the corresponding changes that we have made to the manuscript. In the revised manuscript, in addition to addressing these comments, we have also taken the opportunity to edit the text to remove unnecessary adjectives and adverbs and simplify some of the statements. We have also changed the title to a more detailed and descriptive one. All these changes are also tracked on the revised manuscript we submit here. We have now also posted a preprint of the revised manuscript on arXiv at <https://arxiv.org/abs/2503.22910>, with the only difference being that the arXiv version has more references given that arXiv does not place a limit on the number of references.

Reviewers' comments:

Reviewer #1 (Remarks to the Author):

1. The manuscript introduces a low cost scanning electron microscope developed by the authors and demonstrates applications.

The manuscript is well written, and can be impactful by making scanning electron microscopy more accessible. The authors make an excellent point that democratizing scientific instrumentation benefits society.

That said, the work presented in the manuscript is not a good match for the Nature Communications in terms of novelty, depth and mainly the topic itself (cheap desktop SEM capable of handling wet samples has been described before.). Therefore, suggest rejecting the manuscript.

Response

We thank the reviewer for evaluating this work. However, we respectfully disagree with the reviewer's assessment. This work is not about introducing a particular low cost scanning electron microscope. Rather, it is to demonstrate that a broad and highly useful regime of the performance-cost-flexibility space can be addressed precisely without the need for a specific (and inevitably complex) design. That is the novelty and what sets the work apart from existing approaches and instruments. This opens up a new path compared to what is possible in the world of traditional electron microscopy, and allows non-specialists to create simple instruments to satisfy currently unaddressed needs. We think of it as the electron microscopy equivalent to the magnifying glass.

Our instrument costs roughly an order of magnitude less than the "cheap" SEMs that the reviewer has referred to. In terms of accessibility and flexibility, this manuscript empowers non-experts to build their own instruments and integrate them into their setups and experimental workflows, in ways that existing desktop instruments simply do not allow.

2. Moreover, the manuscript does not provide sufficient information (blueprints, electronics parts list and source code for of their software) to reproduce their implementation of low-cost SEM decreasing the usefulness of the current version of the manuscript.

Response

We fully agree that it will be useful to make the complete detailed design of our specific implementation of this concept available for others to replicate. Indeed, as already indicated in the "Data availability section" of the original manuscript (now updated and on line 482 with a link to detailed design files), we had planned for those to be made available online together with the publication of the manuscript.

However, the crucial point is that the information needed to make this instrument is actually already included in the manuscript and Methods. For example, where we said a laser pointer with a power of a few tens of milliwatts, we literally meant that any such pointer would work—there is no need for a specific laser product, nor a particular wavelength or polarization (although we had specified the wavelength we used). The same goes for the power supply, magnets, electron-optical column, etc., where we have already given the parameters that matter (voltage, ripple, current, magnetic field, dimensions, required tolerance, materials) but, beyond those, the specifics of the design and chosen components are completely open to choice.

Figures 1a and 3a demonstrate the essence of the architecture (with actual photos of the components used, as mentioned in the manuscript) and, together with the text mostly in the Methods section, the full electron-optics design (with actual numbers, dimensions, and consequential component parameters). Specifically, from the original manuscript, we highlight the accompanying descriptions of Figure 3 (now starting on line 173), and all the relevant details including the parameters and dimensions then given in the sections “Optically excited thermionic electron source based on CNT forest” (now line 280), “Cathode assembly” (now line 301), “Analysis of electron-optical demagnification, aberrations, and probe size” (now line 337), “Point-projection measurements of the crossover and probe” (now line 363), and “Beam column” (now line 398).

In other words, others do not have to follow our specific embodiment (structure, form factor, components used, etc.), and can create diverse implementations that suit their own needs, based on information already provided in the manuscript. Making the paper about the blueprints of our specific implementation would thus defeat the purpose. However, as mentioned above and already stated in the original manuscript, our detailed blueprints, etc. will also be provided separately online to complement the manuscript.

We have now further clarified this starting on line 75, and provided an online link to the complete blueprints in the revised manuscript on line 483. (This link is currently private for *Nature Communications* review purposes.)

3. A more detailed feedback follows below.

While I'm highly supportive of open science and in particular new open hardware for electron microscopy, I believe the manuscript is better suited for publication in a microscopy and / or hardware oriented journal. Example journals could be:

Hardware-X

<https://www.hardware-x.com/content/aims>

or

Methods in Microscopy

<https://www.degruyter.com/journal/key/mim/html?lang=en>

or

Micron

<https://www.sciencedirect.com/journal/micron>

Response

Those venues would have been suitable had the manuscript's focus been on new hardware or a particular SEM design following existing approaches but, as mentioned in our response above, that is not the case.

We believe that it is important to take a step back, consider what has been achieved (the images shown / performance demonstrated), and the great simplicity behind it as represented in Figures 1a and 3a. This is showing that you can point a laser pointer to a nanomaterial to excite an electron beam that is adequate for focusing by only an everyday ring magnet and detecting by a generic photodiode in order to produce useful images (while the support structure and

steering/control electronics are open to choice and need only a basic shop to implement). Nothing of this nature has been demonstrated before; this is more of a discovery-invention and demonstration of a new possibility, rather than the specific implementation of a new piece of hardware.

Also, importantly, the target audience of this work is extremely broad and primarily beyond the electron microscopy community.

Therefore, it is important to have this work published in a multidisciplinary venue that can both highlight its novelty and help establish the scientific credibility that it deserves, and at the same time help it rapidly reach broad and diverse audiences and communities. We thus believe that the work is highly synergistic with *Nature Communications*.

4. To make their argument on democratizing electron microscopy solid, the authors should publish the blueprints and all the code for their set-up in an accessible manner. The above-mentioned journals as well as Open Science Foundation, GitHub or GitLab are examples of suitable platforms to make the code and blueprints available.

Response

Please see our response to the reviewer's comment #2 above. As we had already indicated in the original manuscript, our own specific implementation will be made available online (together with the publication of the manuscript), and that is still our plan. It will include the detailed CAD files, circuits, codes, and components list.

We have now further clarified this starting on line 75, and provided an online link to the complete blueprints in the revised manuscript on line 483. (This link is currently private for *Nature Communications* review purposes.)

5. Their claim of sub-5000 dollar part list is misleading. The vacuum pump and controller that they use alone is close to 10,000 dollars.

Response

We quote from the original manuscript (now modified and on line 155): "The bill of materials for our instrument, *not including the vacuum pump (more on that below)*, was ~\$5,000 US ..." Later in the original manuscript, we had an entire section "IV. Vacuum considerations" (now line 201), where we provided the vacuum levels involved (1e-2 Torr and 1e-4 Torr), which clearly do not require an ultra-high-vacuum nor even a high-vacuum pump. We then had another section "Pumping system used for our prototype" (now line 441), which stated that we used a turbomolecular pump (convenient for systematic studies) and described why such an advanced pump is actually not required.

Furthermore, note that we did not particularly attempt to make our prototype low-cost and, for example, used unnecessarily expensive machining services and vacuum parts, since our goal was to demonstrate the concept and not optimize for cost. We had explained this in the original manuscript and that the cost may still be cut significantly (now line 156). Therefore, even including such a pump would still bring the total cost to the order of \$10k, and that is still about an order of magnitude lower than typical existing "cheap" electron microscopes.

The manuscript had also extensively discussed how the low-vacuum-requirement nature of the source greatly facilitates the development of future pumpless devices.

We have removed the cost estimate from the abstract of the revised manuscript, in order to prevent any misunderstanding.

6. The use of permanent magnet lens has been well established since 1950's. For example, RCA offered a commercial 50 kV TEM utilizing permanent magnet lens.

http://www.smecc.org/rca_emt_tabletop.htm

More recently, permanent magnet excited lens are being used in Hitachi H300 (1980's), and in TM-1000 and TM-2000 SEMs.

Response

We never claimed that using permanent magnets was a novel element of our work. To the contrary, in the original manuscript, we had explicitly said (now line 416): "The use of permanent magnets to focus an electron beam has been studied extensively in the past [13]."

7. Abstract:

The authors say that their design is a flexible design, suitable for customization. To support such claim they should show some flexibility (other than lens positioning) that their design enables.

Response

The point about flexibility is precisely that there are no particular design constraints, other than the basic electron-optics shown and described in the manuscript. The form factor and construction are up to the user. (But of course we provide our own detailed design, as discussed before.) The only things that need to be followed are the basic electron-optics described in the manuscript—the rest is flexible. Please refer to our response to the reviewer's comment #2 above for more detail.

8. Main text:

It is not entirely correct that electron microscopy is not accessible. I'm aware of activities where table-top SEMs are loaned to high-schools free of charge to promote science and microscopy. The program I happen to be aware of has been running close to a decade. So while the authors make it possible to built own SEM, it is not true that SEMs are not accessible.

Response

We disagree with the logic of the reviewer's statement. To the contrary, the reviewer's comment precisely supports our point about inaccessibility: if electron microscopes were broadly accessible, there would not be a need for special loan programs, which would inevitably be limited in scope and availability. The reviewer provides no indication of broad accessibility. We know of many university colleagues in countries who have no access to electron microscopes, some of whom even have to send their samples to other countries for imaging, let alone high-school students. (As a side note, we have also received great interest from high-school educators.) Furthermore, science promotion is but one of a vast array of potential applications of our instrument, most of which require one to have one's own instrument for permanent use and even incorporation into other experimental setups. This is not addressed by occasional loan programs. If a limited group of people have the option to borrow an expensive piece of equipment for limited use, that does not mean that the equipment is broadly accessible to society—exactly the opposite.

9. Projects comparable to the design reported here can be found even on YouTube

<https://youtu.be/VdjYVF4a6iU?si=dgfWAG6QxwaFou7C>

arguably doing a good job democratizing SEM, as pointed out by the authors. In my view the Applied Science YT video provides more useful and detailed practical information than the current manuscript.

Response

We had already cited that work in the original manuscript, as the reviewer has acknowledged. That video/project depicts a standard approach using a traditional electron source. While a highly worthy effort, the instrument built is a large and bulky machine, and does not show high imaging

performance. The images obtained by that project (e.g. see around time stamp 9:58 in that video) do not compare with what we show in the manuscript. That project/video, while a testimony to the extraordinary skills and abilities of the owner of the Applied Science YouTube Channel, precisely serve to highlight the challenge of traditional approaches in this context, and thus the significance of our result and the desire for an approach like what we have demonstrated. In fact, helping equip such skilled enthusiasts with this novel approach and instrument concept is directly in the spirit of our project.

Indeed we are also considering possibly making videos of our implementation in the future to add to our online documentation. However, that would not be a replacement for a research paper, but a complementary activity.

10. In addition to the SEMs mentioned on line 45, both Hitachi and JEOL supply competitive tabletop SEMs.

Response

On that line (now line 51) we had explicitly said that what we have listed are only “examples.” Since this is a research paper, we do not find it relevant to provide a comprehensive list of all commercial tabletop SEMs, in particular given that our work is about a completely different combination of performance, price, and flexibility than all those instruments, and not meant to compete with them.

11. The authors are not entirely correct in their description of the NanoMi project. It provides detailed information, including blueprints, to build pretty much anything: from physicist-instrument to portable hobby-level tools. It is actually rather simple platform to implement in its basic configuration and detailed information, is available either on Git or upon request.

Response

We find project NanoMi exciting and highly valuable. In fact, we believe that there may exist potential opportunities for us to collaborate with the NanoMi team in the future.

At the same time, it is important to refer to NanoMi in an accurate manner. The reviewer’s statement that NanoMi can be used “to build pretty much anything” is not supported by the NanoMi literature. NanoMi follows an essentially traditional approach to electron microscope design based on established concepts, with its design being made openly available. That is a highly worthy effort, but the reviewer’s claims about simplicity and comparison to our work are not justified. NanoMi represents a completely different approach, regime of complexity, and target performance compared to our work. Each has its own place and value, and suggesting that NanoMi takes away from the novelty and impact of our work is factually unsupported and highly unfair.

We have nonetheless reworded our statement about NanoMi to highlight its promise (now line 57). While, due to the limitations on the number of references allowed by *Nature Communications*, we have cited only the NanoMi journal paper, in the arXiv version of the manuscript (<https://arxiv.org/abs/2503.22910>) we have also included numerous citations to NanoMi conference papers and the NanoMi website.

12. The authors point the large depth of field in their instrument. However, that feature is inherent to SEM imaging at modest resolution. The application images in Figure 1 are a good choice demonstrating the utility of an low-cost instrument.

Response

We thank the reviewer for recognizing the value of the images in figure 1. We had not claimed that the large depth of field was a unique feature of our instrument. To the contrary, the opening line of

the abstract (now line 8) and the second sentence of the introduction (now line 30) had both mentioned that that is a feature of the SEM in general. We had simply pointed out the important fact that our instrument, despite its simplicity, achieves a usefully large depth of field—a property that plays a critical role in the image quality.

13. The use of laser-heated thermal source is an excellent idea that could be of interest to open-source electron microscopy community due to its practical advantages. The authors have built two instruments, demonstrating the repeatability of their design. It would be desirable to also demonstrate that their design can be also reproduced at other labs.

Response

We thank the reviewer for highlighting this crucial point about the electron source. We fully agree and indeed hope to reach out to other groups for collaborations once the paper is published. In fact, already several colleagues and laboratories (both locally and internationally) have expressed a strong interest to build versions of the instrument.

14. On line 154 the authors state:

"Chromatic and spherical aberrations are calculated at 32 nm and 3 nm, respectively"
do they mean chromatic and spherical aberration coefficients ? Also check Figure 3 for units.
If so, the units are likely millimetres, not nanometers.

Response:

No, the values given were indeed the actual probe size contributions from the aberrations, and are in nanometers. This had been described in the section "Analysis of electron-optical demagnification, aberrations, and probe size," specifically starting from (now) line 346 where the calculations of the aberrations were presented based on equations 1 and 2. Therefore, the values and units given in the text and Figure 3 (nanometers) are correct.

15. The authors claim that their instrument cost is ~ 5 kU\$. Does it include vacuum pumps ? The text does not seem to specify the type of pumps used, but that is important information.

Response

The Methods had a section "Pumping system used for our prototype" (now line 441), which had mentioned that we used a turbomolecular pump (for convenience for systematic experiments) and also described why such an advanced pump is actually not a requirement. The manuscript also had a section "IV. Vacuum considerations" (now line 201), where we provided the vacuum levels involved ($1e-2$ Torr and $1e-4$ Torr), which clearly do not require an ultra-high-vacuum or high-vacuum pump. We had also explicitly stated on (now) line 155 that the \$5k cost did not include the vacuum pump. Please refer to our response to the reviewer's comment #5 for more detail.

16. The summary and outlook section is largely speculative and not supported by results

"Multiple independent detectors may be placed around the objective aperture to cover different backscattering angular ranges, and their signals used for enhanced topographical imaging. "" has not been demonstrated here.

"In addition to backscattered electron detection, other established imaging modalities may be used, including electron-beam-induced current and secondary electron detection using simple gaseous detectors. Additional electron-optics may also be incorporated to provide magnification

after the specimen for transmission-mode imaging." While all of this is plausible, it was not demonstrated and can not be claimed.

"The addition of X-ray or cathodoluminescence detectors would also enable spectroscopy." As above, when not demonstrated it should not be claimed in a journal with level of Nature Communications.

While AI is an important direction in electron microscopy, there is no link, beyond speculative suggestions, between developments and the information in the manuscript.

Response

We had included this separate section to paint a picture for what may be done in the future by the community. We had not made any claims about those, and instead described them as part of an "Outlook" and therefore future possibilities ("... may be placed ...," "... may be used ...," etc.). What has been demonstrated in the manuscript is quite clear. We think, with a novel path such as this, and especially given that this manuscript is intended for a broad audience beyond electron microscopists, it is important to provide some inspiration and guidance for future work.

17. --- Methods:

The use of laser-illuminated CNT forest as a thermal emitter is arguably a novel feature. In any case, it is an important aspect making the instrument robust to poor vacuum. As such, the authors should provide sufficient detail to enable others easily reproduce their set-up. For example, the (commercial) source of CNT samples, the model of the laser(s) tested etc. should all be provided.

Response

We thank the reviewer for recognizing this novelty. We had described our nanotube growth process in the Supplementary Information (referenced in the main text—now line 119) section "Carbon nanotube forest growth and characterization." Commercial sources, laser models, etc. can be chosen from among various options. Our specific component choices were intended to be provided as part of the online design documentation together with the publication of the paper, as had been stated in the original manuscript. Please refer to our response to the reviewer's comment #2 for further detail.

We have now further clarified this starting on line 75, and provided an online link to the complete blueprints in the revised manuscript on line 483. (This link is currently private for *Nature Communications* review purposes.)

18. The authors should provide a plot of electron beam current versus time and possibly illumination power to support their claim of the electron current provided by the source and of its stability.

Response

We have published on this source in the past and those relevant past publications had been referenced in the original manuscript (references [1] for a review, as well as [24-31, 37, 57, 58, 61] for details on various aspects of the physics and properties of this heating effect and electron emission). Repeating any of those here is beyond the scope of the present work. The images included in the original manuscript clearly demonstrate the source's sufficient current and stability for this electron microscopy application. The original manuscript had also described the laser wavelength and power we used (now line 288), which are the only important parameters for this application.

19. "... " A set of 4 small, 450-turn off-the-shelf ... "
and

"The objective lens consists of two off-the-shelf, axially-magnetized neodymium ring"
The authors need to specify in more detail what they used. When they mention "off the shelf" component, it is necessary to include manufacturer and model.

Response

As mentioned before, we intended for all this information related to our specific implementation to be made available online together with the publication of the manuscript.

We have now further clarified this starting on line 75, and provided an online link to the complete blueprints in the revised manuscript on line 483. (This link is currently private for *Nature Communications* review purposes.)

However, most importantly, please refer to our response to the reviewer's comment #2 above. Also, as the original manuscript indicated, the photos of the specific components are also seen on Figure 1, so that the reader can gain a feel for them. But truly, again, the specific components we used are not mandatory. For example, we have given both the field distribution of the ring magnet (-0.25 T, see figure M2) and its diameters (now line 415), and those are what matters.

20. Values of figures such as Fig 3 is nearly zero (they exist in nearly all textbooks of electron microscopy) unless they provide detail of a particular implementation, such as dimensions, coil type, exciting currents etc.

Response

It appears that the reviewer has misinterpreted Figure 3 as a generic schematic, but that is not the case. Figure 3a describes the actual electron-optics of our instrument, and Figure 3b shows specific point-projection microscopy results that corroborate the imaging resolution of the instrument; both are entirely crucial. The accompanying descriptions start on (now) line 173, and all the relevant details including the parameters and dimensions had already been given in the sections "Optically excited thermionic electron source based on CNT forest" (now line 280), "Cathode assembly" (now line 301), "Analysis of electron-optical demagnification, aberrations, and probe size" (now line 337), "Point-projection measurements of the crossover and probe" (now line 363), and "Beam column" (now line 398).

We believe that this comment by the reviewer further highlights the extreme simplicity of our instrument and its vastly different nature to existing electron microscopes, to the point that the reviewer appears to not have appreciated the fact that Figure 3 depicts (together with the text referred to above) the actual electron-optical design of our instrument, and instead mistaken it for a generic diagram.

21. Paragraph "electronics" is useless. There is no information beyond the irrelevant fact that the authors like Teensy microcontroller and do not need much power.

Response

We believe that low power is important for many applications. In the revised manuscript, we have made that paragraph more concise (now line 427) and removed the name of the microcontroller, as that is more aligned with the rest of the manuscript not emphasizing specific components.

22. As for the vacuum section, the controller and TMP the authors use costs in the order of 10,000 dollars. This implies that their stated 5,000 dollar instrument cost is misleading. Furthermore, even with Quik Flange connectors and correct (e.g. Torr Seal) epoxy and multiple moving parts, the authors should be able to routinely achieve 10^{-6} torr or less.

Response

The point we had made was that our instrument does not need high vacuum, and that is very important. Regarding the pump and price, please refer to our answer to the reviewer's comments #5 and #15.

Furthermore, in keeping with the rest of the manuscript and not emphasizing specific components, we have now removed the pump make/model from the revised manuscript and now only mention that it was a turbomolecular pump (line 442).

23. In the supplement. info the authors again refrain from including useful information, such as thickness and shape of mu metal shielding, type and dimensions of magnets used for objective lens etc. While I do agree that the optics performance is and choice of magnets is not critical, the authors need to include such information to make it possible for others to reproduce their design.

Response

Please refer to our response to the reviewer's comment #2. Also, the type, strength, and dimensions of the magnets had already been given (manuscript Figure M2, and Supplementary Information Figure S4 labels). The mu metal shielding had only been shown as Supplementary Information and, as originally stated, not even used for the images shown in the manuscript. In any case, we again reiterate that all of the details of our specific implementation were to be provided online together with the publication of the paper, as had already been stated in the original manuscript.

We have now further clarified this starting on line 75, and provided an online link to the complete blueprints in the revised manuscript on line 483. (This link is currently private for *Nature Communications* review purposes.)

Reviewer #2 (Remarks to the Author):

The article presents an innovative method for making a scanning electron microscope from simple, inexpensive and available materials and elements. According to the authors, the price of the components and manufacturing techniques used does not exceed USD 5,000 (excluding the vacuum pump), which is an impressive result when it comes to the price of the manufactured device. The presented operating parameters of the microscope are not as good as those of its expensive counterparts, but competing with high-tech devices, whose design has been developed for decades, is basically impossible. The image resolution presented in the article is much better than that of the best optical microscopes. This was the goal of the creators, to provide users with an electron microscope that is cheap to manufacture and easy to use, and at the same time its imaging capabilities are below 1 μm , thanks to which it can be successfully used to diagnose sub-micrometer structures without additional preparation. This is detailed in the manuscript. The possibility of using the microscope without complicated pumping systems and in an average vacuum is an additional advantage that reduces the costs of such a device (although the purchase of the turbomolecular pump used in the presented set is a large expense), especially since the authors show that it is possible to operate such a device without a vacuum pump, as a tightly closed vacuum device, but this requires additional research and technological work.

The presented work is very important for the field of vacuum and electron beam techniques, such as scanning electron microscopy. Until now, the development of modern microscopic techniques has taken place in closed laboratories of companies that produce these devices. The demonstrated invention provides tools for the development of scanning electron microscopy in the comfort of your home or in any laboratory that would like to have an electron microscope. The authors describe in detail the steps needed to produce this device and also show the latest trends in the development of electron microscopy, in the form of work leading to the miniaturization of

electron microscopes or their use for observing samples placed in atmospheric pressure. They list the possibilities of developing their invention, which can be implemented by other research teams and possibly also by the creators themselves. The presented work shows that the development of even advanced devices can be carried out in any laboratory and that there is still a need for innovations in the field of science and technology.

The conclusions of this article are supported by many images of different samples taken in the developed microscope, as well as by detailed analyses of the electron path in the developed electron-optical column. The possibilities of focusing the electron beam using a permanent magnet and further possibilities of developing the electron-optical column are shown. The work of the carbon nanotube emitter is discussed, and sources where more detailed information about the electron source can be found are given. The article explains that the work of such a source does not require high or ultra-high vacuum, and its operation time is very long, which also confirms the usefulness of the technology used for its invention.

All data and analysis are insightful and appear to be accurate. The analysis is well thought out, and the results are presented clearly. No changes are required in this area.

The article presents the production of a scanning electron microscope under home conditions without the need for knowledge of advanced engineering techniques. For this reason, the presented methodology is simplified to the minimum and is described in such a way that it is possible to produce a similar device on your own. The authors themselves produced a forest of carbon nanotubes that served as a source of electrons, which would be difficult to do at home. However, they write that such structures are available for sale, but searching the Internet, it is difficult to find examples of companies selling such products, so it would be worth adding a link to an example company to confirm that such products are actually available. In summary, this article may be a breakthrough in the availability of scanning electron microscopes for scientists and people who want to use the benefits of these machines in their research. Since I did not find any major flaws in the article, I suggest that it be approved for publication.

Response

We are truly grateful to reviewer 2 for this positive and encouraging assessment. It is refreshing to see that reviewer 2 has fully appreciated our message and recognized exactly what this work is about.

Links to commercial sources of nanotube forests are included in the detailed implementation documentation (CAD, circuits, codes, components list), which has now been linked on line 483 of the revised manuscript. We also provide two examples of possible commercial sources for carbon nanotube forests here:

<https://www.cheaptubes.com/product/multi-walled-carbon-nanotube-arrays/>

<https://www.nano-lab.com/alignedcarbonnanotubearrays2.html>

We grow our own nanotube forests and can also supply them to a degree to others interested in building their own instruments. As the community grows and demand increases, we will work on finding sustainable ways of providing this material to those who do not have access otherwise. The growth process is scalable, so the material can be provided to the community at a low cost, depending on the circumstances.

Reviewer #3 (Remarks to the Author):

Congratulations to the authors on their fantastic expectations to have a Scanning Electron Microscope (SEM) at a cost of around a thousand dollars. However, it seems that in my opinion it is too greedy to have a sealed column SEM with 25 to 35KeV electron beam energy at that cost and the mentioned column size. I propose that you can attempt to have an SEM with a beam energy between 1 to 5KeV at a cost as you wish. Please avoid using aluminum apertures.

Response

We are grateful to the reviewer for recognizing the importance of this work.

The sealed-off column that we have already implemented (Supplementary Information, page 3) actually operated at 30 keV. We have now added this information at the top of page 4 of Supplementary Information. We obtained better detection sensitivity at 25-35 keV beam energies, and those higher beam energies also allow for better transmission through electron-transparent membranes. But we fully agree with the reviewer that lower beam energies are also important, and we intend to also explore those further in the future.

Our objective aperture is made of molybdenum. This has been added on line 406 of the revised manuscript. For the anode plate, which has a wide aperture (1 mm in diameter) and is less sensitive, we used aluminum due to cost and availability considerations, and it appears to work well for that purpose.